# Finding Suitable Transect Spacing and Sampling Designs for Accurate Soil ECa Mapping from EM38-MK2

**Hugo M. Rodrigues** [1], **Gustavo M. Vasques** [2,*], **Ronaldo P. Oliveira** [2], **Sílvio R. L. Tavares** [2], **Marcos B. Ceddia** [1] and **Luís C. Hernani** [2]

1 Departamento de Solos, Universidade Federal Rural do Rio de Janeiro, Rodovia BR 465, Km 07, S/N, Seropédica, RJ 23890-000, Brazil; rodrigues.machado.hugo@gmail.com (H.M.R.); marcosceddia@gmail.com (M.B.C.)

2 Embrapa Solos, Rua Jardim Botânico 1024, Rio de Janeiro, RJ 22460-000, Brazil; ronaldo.oliveira@embrapa.br (R.P.O.); silvio.tavares@embrapa.br (S.R.L.T.); luis.hernani@embrapa.br (L.C.H.)

\* Correspondence: gustavo.vasques@embrapa.br; Tel.: +55-21-2179-4565

**Abstract:** Finding an ideal sampling design is a crucial stage in detailed soil mapping to assure reasonable accuracy of resulting soil property maps. This study aimed to evaluate the influence of sampling designs and sample sizes on the quality of soil apparent electrical conductivity (ECa) maps from an electromagnetic sensor survey. Twenty-six (26) parallel transects were gathered in a 72-ha plot in Southeastern Brazil. Soil ECa measurements using an on-the-go electromagnetic induction sensor were taken every second using sensor vertical orientation. Two approaches were used to reduce the sample size and simulate kriging interpolations of soil ECa. Firstly, the number of transect lines was reduced by increasing the distance between them; thus, 26 transects with 40 m spacing; 13 with 80 m; 7 with 150 m; and 4 with 300 m. Secondly, random point selection and Douglas-Peucker algorithms were used to derive four reduced datasets by removing 25, 50, 75, and 95% of the points from the ECa survey dataset. Soil ECa was interpolated at 5 m output spatial resolution using ordinary kriging and the four datasets from each simulation (a total of twelve datasets). Map uncertainty was assessed by root mean square error and mean error metrics from 400 random samples previously selected for external map validation. Maps were evaluated on their uncertainty and spatial structure of variation. The transect elimination approach showed that maps produced with transect spacing up to 150 m could preserve the spatial structure of ECa variations. Douglas-Peucker results showed lower nugget values than random point simulations for all selected sample densities, except for a 95% point reduction. The soil ECa maps derived from the 75% reduced dataset (by random sampling or Douglas-Peucker) or from 13 transect lines (80 m spacing) showed reasonable accuracy (RMSE of validation circa 0.7) relative to the map interpolated from all survey points (RMSE of 0.5), suggesting that transect spacing of 80 m and reading intervals greater than one second can be used for improving the efficiency of on-the-go soil ECa surveys.

**Keywords:** proximal soil sensing; geostatistics; kriging; uncertainty

## 1. Introduction

Sampling design is fundamental in research and monitoring of natural resources. Proximal soil sensing (PSS) technology is currently available to produce soil attribute maps in high spatial resolution, aiming to support sustainable variable rate input management in precision agriculture [1,2]. However, optimal sampling designs using continuous PSS surveys are still lacking the definition

of operational standards, potentially compromising map uncertainty evaluations. Along with the spatial distribution due to the distance between survey track lines, the sample density in each transect line may affect output map accuracy. These are fundamental parameters in sampling design for detailed soil attribute mapping that can affect efficient use of the so-called on-the-go PSS technology [3]. Although optimal transect spacing frameworks for soil sensing are not new [4–6], recent works show that methodological research is still need on customized approaches according to specific landscape, crop type, soil management, survey strategy, sensor type, and target variable [7,8].

Conventional techniques to obtain soil data samples are laborious from field collection to laboratory analysis, which may constrain sampling design and the consequent mapping uncertainties. To provide proper spatial and temporal scales required for within-field variability analysis, PSS technology has been successfully used for management zone delineation and variable rate applications. PSS is described as using field-based sensors to obtain signals from the ground when the sensor's detector is in contact or close to (within 2 m) the soil [9].

Similar to sensors mounted in different platforms, such as satellites or aircraft, PSS technology is offered in a wide variety of models and sensing methods regarding operational protocols and physical-chemical principles involved in data being collected and analyzed. One methodology getting widely used in agriculture involves commercial devices to monitor soil apparent electrical conductivity (ECa) by contact or by means of electromagnetic induction (EMI). Over several decades, numerous EMI sensors have been used in soil science research and applied precision agriculture solutions. Popular commercial devices include models from VERIS® (Veris Technologies, Salina, KS, USA), DUALEM (DUALEM Inc., Milton, ON, Canada) and Geonics (Geonics Limited, Mississauga, ON, Canada). All of these companies provide multiple models for different applications as Geonics EM31, EM34, EM38, EM38DD, and EM38-MK2.

Soil ECa mapping has been recognized as a promising method in precision agriculture [6] to measure the spatial variability of soil properties at field and landscape scales [8,9]. Electromagnetic induction (EMI) sensors measure soil ECa based on the behavior of soil chemical elements [10–14]. EMI devices do not require direct ground contact, so data collection is relatively easier, faster, and less invasive when compared with contact models. This technology is particularly suitable for on-the-go intensive soil monitoring operations, providing a significant number of sample points and more comprehensive area coverage than would be feasible using traditional core-sampling methods and laboratory analysis.

EMI devices use active ground conductivity meters (GCM) sensors, consisting of transmitting and receiving coils producing alternating electrical currents in the ground. Electrical currents start in the transmitting coil and generate a primary time-varying electromagnetic field. This primary field induces eddy currents to flow through the soil, thus creating a secondary electromagnetic field due to the soil electrical conductivity properties. Resulting electrical-current amplitude and phase generated by the secondary electromagnetic field are measured by the receiving coil [15]. Under conditions known as "low induction number operation" [16], the secondary electromagnetic field is proportional to earth induced currents, and can be used to calculate soil ECa, commonly expressed in units of millisiemens per meter (mS/m) [15].

A popular PSS device to measure soil ECa is the EM38 (including its sub-models, EM38-MK2 and EM38DD), which have been used in several applications. Researchers have used some version of EM38 to monitor soil salinity and relate it to laboratory analysis from the beginning [17–21]. Using the principles of data fusion, some studies have used DUALEM-1S and DUALEM-421S to measure soil water concentrations [22–24]. Veris® MSP and 3100 instruments, as well as EM38 and EM31 have been used to measure texture [25–27]. Dualem-421S and EM38DD have been used for mapping cation exchange capacity [28,29]. More recent applications have used EM38, EM34 and EM31, Veris® 3100 and KT-5 (SatisGeo, Brno, Czech Republic) for mapping with-in-field spatial variation of soil types [30–32]. Soil mapping in 3D has been studied with Dualem-421S [28] while EM38-MK2 has been employed

in tailored data acquisition structures [33]. Therefore, large georeferenced datasets are gathered by continuous recording during less invasive and high-speed operations.

High density data sampling can provide an efficient characterization of soil property variations [32,33]. Sudduth et al. [34] recorded more than 5000 observations, corresponding to a 4–6 m data spacing, and Islam et al. [35] recorded more than 14,000 observations in a 1.4 ha ECa dataset. They applied the EM38-MK2 on a 1 s interval, stating that amount of points could enable automatic variogram fitting, provide proper kriging interpolations, optimal-design, and fast-navigation paths as required for cost-effective survey interventions. Whereas high density datasets may reduce bias in sampling designs [25], sample density directly affects output map uncertainties. Therefore, studies to tailor transect spacings and sample densities should be considered to overcome soil ECa map uncertainties when integrating mobile visualization (on-the-fly) and on-the-go monitoring for variable rate application decision support. In this context, this work aims to contribute to these questions, establishing efficiency thresholds to maintain output map accuracy. Investigations on different sample designs look to the reduction of transect lines and sampling observations as a matter of minimum track distance and maximum operational speed, respectively. The overall objective is to evaluate interpolated on-the-go ECa maps from different sample designs using common validation indices. It is believed that a proper combination of sampling designs can improve operational efficiency, preserving high quality with low uncertainty in map generation. The paper introduces preliminary analysis of continuous EM38 operational frameworks in Brazil, and it could provide basic information on optimal PSS sampling designs in tropical-soils that are relevant to central pivot no-till grain production.

## 2. Materials and Methods

This section details materials and methods related to soil ECa survey dataset investigations on sample distribution and density influencing output map accuracy. Specific objectives are addressed by considering: four transect spacing subsets; four sampling density subsets using the random point and four using the Douglas-Peucker selection algorithms; and kriging interpolations evaluated by a standard external validation subset for mean error (ME) and root mean square error (RMSE) indexes.

### 2.1. Study Area

The study was carried on a grain crop rotation production system (i.e., beans, soybeans, wheat, and oats) under central pivot irrigation and no-till soil management. The farm is located at Itaí district, São Paulo State, Brazil (Figure 1a). It has central coordinates of 23.58544° South latitude and 48.9395° West longitude, in a subtropical climate, with annual average maximum and minimum temperatures of 26 °C and 16 °C, respectively, and an average rainfall of 119 mm [36]. The paddock area is 72 ha at a maximum elevation of approximately 712 m above sea level (Figure 1b). The regional soil characterization is an association of LATOSSOLOS VERMELHOS Distróficos ("*Ferralsols*") and ARGISSOLOS VERMELHOS-AMARELOS Distróficos ("*Acrisols*") (Brazilian Institute of Geography and Statistics, 1:5000,000), The pivot area was offseason with wheat straw cover during survey.



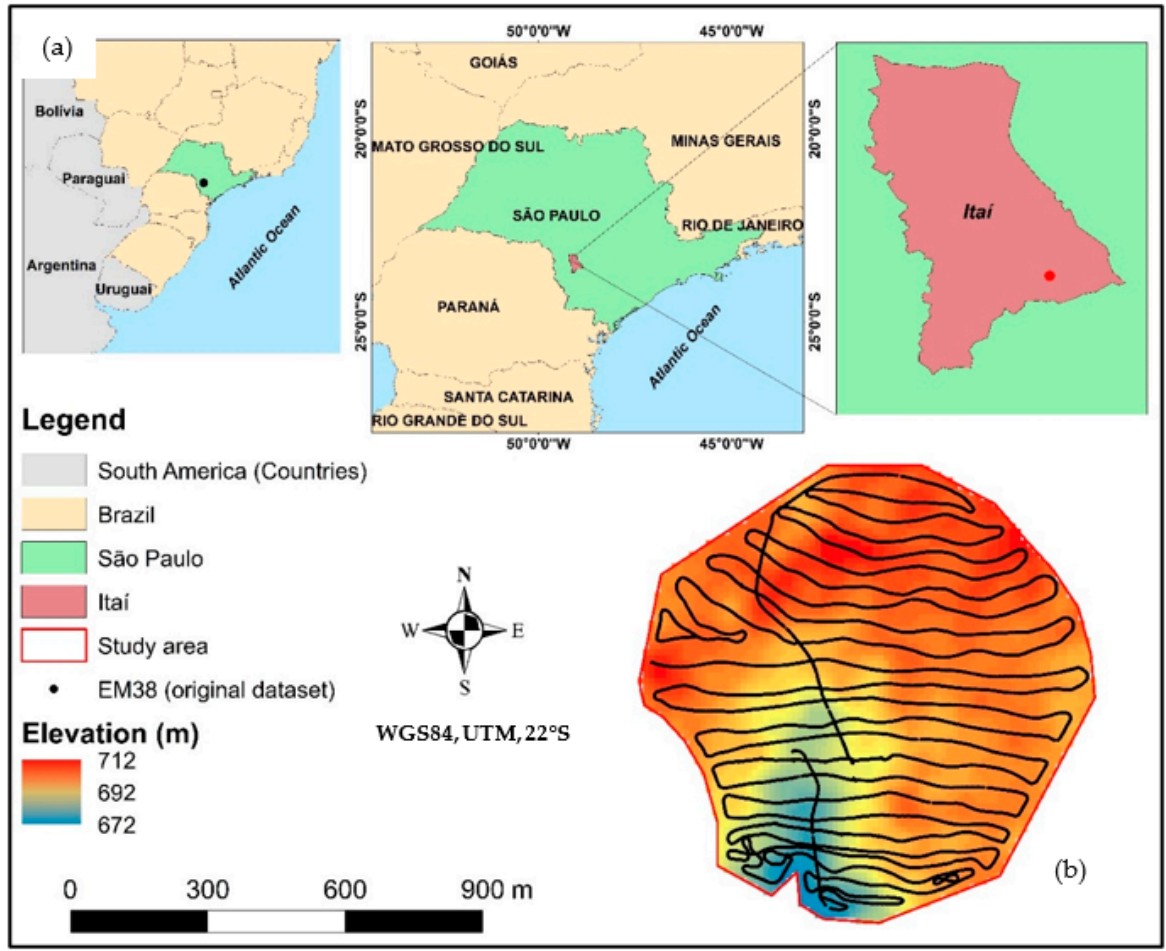

**Figure 1.** Maps showing (**a**) study area location and (**b**) navigation path over elevation.

*2.2. EM38-MK2: Mobile Data Acquisition Structure and Survey Operation*

This ECa survey used the EM38-MK2 EMI meter which includes two receiver coils, separated by 1 m and 0.5 m from a single transmitter coil. This device provides two simultaneous ECa datasets with readings in milliSiemens per meter (mS/m), either in vertical or horizontal dipole orientations. The effective depth ranges of ECa readings are 1.5 m and 0.75 m in the vertical position (ECaV), or 0.75 m and 0.375 m in the horizontal position (ECaH).

Field operations started with sensor calibrations at the height of 1.5 m from the ground in both horizontal and vertical dipole orientations. After calibration procedures the sensor was placed perpendicularly to the earth's surface in a mobile data acquisition system as further detailed, providing vertical dipole readings (ECaV). As no measurements were taken in the horizontal dipole orientation, the resulting dataset is further refereed as ECa.

Data storage was done in a single and continuous run, using Bluetooth connections between an Archer Rugged Handheld (Juniper System Inc., Bromsgrove, UK) PDA and two professional georeferencing devices, a XGPS-100A (Dual Electronics Corporation, Lake Mary, FL, USA) roof top GPS and a GeoExplorer 3000 (Trimble Inc., Sunnyvale, CA, USA)

A mobile data acquisition system was structured in a wooden box with no metal parts used. The box was assembled using wood-glue and Velcro tapes wrapping it over a high-resistance rubber mat (1 cm thick). The rubber mat was attached to long nylon straps connected to the back of a 4 × 4 pickup, dragging the structure 3 m apart to avoid magnetic interference from the metallic body (Figure 2).

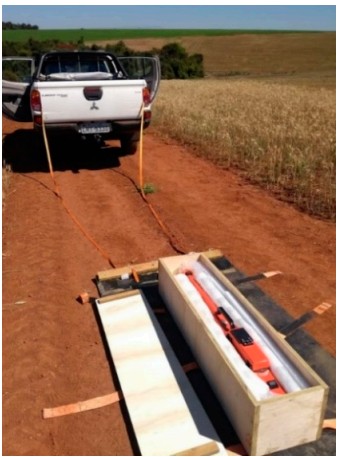

**Figure 2.** Mobile data acquisition structure for EM38-MK2 on-the-go survey.

Navigation speed through the entire study area was kept constant at 15 km/h, taking 90 min for the total operation and collecting parallel transect lines in a back-and-forth path. The average distance between transect lines was approximately 40 m (Figure 1b). The soil ECa survey raw dataset was of 5788 observation points in total. The navigation path included 26 transect lines with the EM38-MK2 sensor set for a 1 sec reading interval.

### 2.3. EM38-MK2 Data Filtering and External Validation

Exploratory data analysis was applied to the EM38-MK2 raw dataset to investigate for outlier values due to potential electromagnetic interferences by metallic parts of the irrigation pivot framework creating high conductivity at specific locations. Spatial query filters were used to remove ECa observations recursively measured in the same location when brief stops for operational maintenance were necessary. Complementarily, sample points that drifted off transect lines were removed to improve a parallel sampling design path. The remaining clean dataset was of 4306 points in total. The final preprocessing step used an automatic random subset sampling algorithm, in the R statistical packages [37], to subset points, reserving 400 points for use in external map validation of the simulations from the different sampling designs (Figure 3).

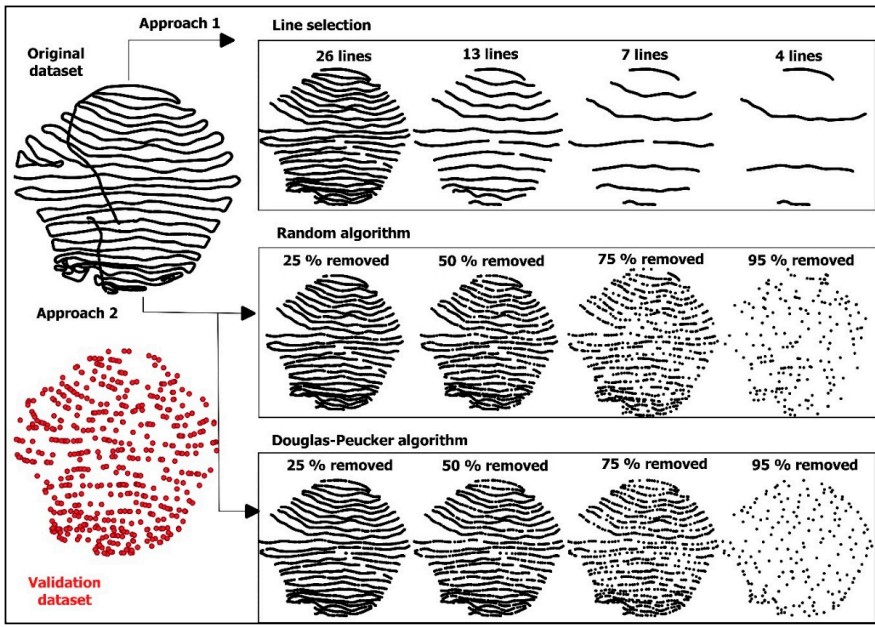

**Figure 3.** Workflow for different sampling designs for the selection of simulation datasets.

*2.4. Sampling Designs*

2.4.1. Approach 1—Different Transect Spacings

　　The clean dataset of 4306 ECa samples in 26 transect lines was used to generate another three sample subsets by increasing transect lines distances and consequently reducing the total number of parallel lines in each subsequent set. From the most detailed dataset of 26 transects, approximately 40 m apart, with 3906 points after external validation subset extraction (i.e., $4306 - 400 = 3906$); the other three simulation datasets used transect distances of 80 m, 150 m, and 300 m, respectively (Figure 3). Table 1 details all four resulting simulation datasets with their respective numbers of transect lines and the remaining dataset size.

**Table 1.** Soil electrical conductivity (ECa) input simulation datasets from all sampling designs, showing: (a) average transect spacings, number of remaining transect lines and dataset size from the transect spacing approach; and (b) sample density and dataset size from random sampling and Douglas-Peucker reduction.

| Varying Transect Spacing | | |
|:---:|:---:|:---:|
| **Transect Spacing (m)** | **Number of Lines** | **Dataset Size** |
| 40 | 26 | 3906 |
| 80 | 13 | 2119 |
| 150 | 7 | 1088 |
| 300 | 4 | 558 |
| Varying Sample Densities | | |
| **Density reduction (%)** | Random | Douglas-Peucker |
|  | Dataset size | |
| 25 | 2930 | 2933 |
| 50 | 1953 | 1960 |
| 75 | 977 | 982 |
| 95 | 196 | 196 |

2.4.2. Approach 2—Different Sample Densities Using Random and Douglas-Peucker Algorithm

　　From the standard survey dataset (3906 points), another four simulation subsets were extracted for sampling designs using different sample densities. The first algorithm used was the automatic random subset sampling algorithm from the R statistical packages [37], which eliminated 25%, 50%, 75%, and 95% of sample points from the original dataset (Table 1 and Figure 3). The same removal percentages were performed by the *DouglasPeuckerNbPoints* function implemented in the *kmlShape* package [38]. This algorithm consists of a proximity rule, where all original data points must be within a certain distance from the estimate. A polyline is created using the input dataset coordinates as polyline vertexes, from which a tolerance distance or an idealized number of points can be predefined. The algorithm strategy recursively creates new segments approximating the original polyline, until all vertices of the polyline satisfy the predefined tolerance condition [39]. Both sampling design approaches were further evaluated to assess kriging interpolation accuracy metrics using an external validation subset as further detailed in the next section.

*2.5. Statistics, Interpolation, and Mapping Uncertainties*

　　Kriging has been used for many decades for spatial interpolation [40] and is one of the geostatistical tools widely used with good references [41]. Ordinary kriging uses only one variable and is one of the most robust and widely used types of kriging [42]. The main objective of kriging is to estimate the value of a random variable, $Z$, where it was not measured.

　　The study considers two assumptions for sampling and geostatistics: (1) sampling by transects to represent the variation in two dimensions; (2) irregular sampling in two dimensions [42]. However,

here, we will briefly summarize some necessary assumptions and equations. The spatial variability of ECa for each group, selected by approach 1 or 2, was analyzed using variograms. In this analysis, the spatial dependence of an observation for a given point *z(x)* is comparatively determined for a specific observation given its neighboring points *z (x + h)*, where *h* denotes the distance lag and *N(h)* is the number of data pairs separated by a particular lag vector *h*. The average distance calculated for each gap of the variogram is given by *γ(h)* [41] Equation (1).

$$\gamma(h) = \frac{1}{2N(h)} \sum_{i=1}^{N(h)} [\{Z_i(x_i) - Z_i(x_i + h)\}^2] \tag{1}$$

The result of the experimental variogram is the mean of the semivariance of the pairs of points $Z_i(x_i)$ and, $Z_i(x_i + h)$, sampled over a lag distance *h*. A variogram model can be fitted to the experimental variogram. Based on the variogram model, values can be estimated at locations that were not sampled using kriging. Moreover, the variogram model provides a value for the nugget variance $C_0$, which is the theoretical semivariance at the sampling location. It is extrapolated from the shape of the variogram model at short lag distances to *h* equal to zero. The nugget variance includes the variance that is associated with the small-scale variability that cannot be further distinguished by the sampling procedure, and it also includes the variability that is caused by analytical and sampling error [43].

When the nugget variance is subtracted from the sill, the structured variance *C* is obtained, i.e., the variance being explainable from neighboring observations. The range (*a*) is the distance at which neighboring observations become spatially independent.

Therefore, to manifest the spatial continuum of observations, the optimum sampling distance must be taken shorter than the range; as observations become less and less related, the more they approach the range. The maximum sampling distance (upper limit of cell size) can be determined with the "mean correlation distance" (MCD) [44], and can be calculated for spherical variogram models with Equation (2).

$$MCD = \frac{3}{8} * \frac{C}{C_0 + C} * a \tag{2}$$

The by-product of ordinary kriging is the kriging variance, and the standard error can be calculated as the square root of the variance. Therefore, this by-product is a spatial variation function of the data (i.e., modeled by the variogram) or the spatial configuration of the data concerning each of its estimated values. The variance of the estimate is the expected value between the $\hat{Z}(x_0)$ and $Z(x_0)$.

Thus, the variance of each map produced from approach two was calculated using the two selection algorithms. Then, to assess the spatial difference between the two methods, each equivalent map in each level of removal density was subtracted from the results of the two different algorithms to identify the spatial variation of the estimated variances.

Statistical analysis of ECa sample subsets for the two approaches was evaluated for normal distribution patterns according to kriging interpolation assumptions. If the data were not normally distributed, they were transformed by the natural logarithm. A manual variogram fitting procedure for all simulations in both approaches used variogram analysis tools from the gstat package [45] in R software [37] for isotropic variogram fitting. Ordinary kriging interpolations used the *krigeTg* function, applied either to normal distribution or natural logarithm transformed subsets.

ECa map accuracy of all combinations was evaluated with an external validation subset using the mean error index (ME) in Equation (3), and the root mean square error (RMSE) in Equation (4).

$$ME = \sum_{i=1}^{N} [y - \hat{y}] / N \tag{3}$$

$$RMSE = \sqrt{\sum_{i=1}^{N} \frac{(y - \hat{y})^2}{N}} \tag{4}$$

where: $N$ is the number of observations, $y$ is the observed value and $\hat{y}$ is the predicted value.

## 3. Results

### 3.1. Approach 1—Different Transect Spacings

#### 3.1.1. Exploratory Data Analysis

Descriptive statistics summarizing soil ECa datasets for different distances between transect lines are presented in Table 2, along with the ECa external validation subset. ECa derived datasets displayed equal minimum and maximum values for all line spacing subsets, except for the 26 lines dataset, which showed a difference in the minimum value (2.62 mS/m). Values for all datasets exhibited mean and median values close to 10 mS/m, indicating that, although transect spacing increased, we can observe similarities regarding mean values. The standard deviation value for the 26 lines dataset was 3.39, similar to the other datasets. Asymmetry values shown in Table 2 can be classified as moderately positive for all datasets. An increasing trend towards the left of the arithmetic mean can be observed in Figure 4a, showing a longer histogram "tail" on the right. Kurtosis coefficients for all spacing datasets, including validation data, can be classified as leptokurtic, as they showed values higher than 0.3, implying that there is a flattening in all data distribution patterns. These asymmetry and kurtosis values suggest that the data distribution for all subsets may not be considered normally distributed. Therefore, a natural logarithm transformation was applied to the datasets to normalize distribution patterns (Figure 4b).

**Table 2.** Descriptive statistics results for soil ECa datasets for the different numbers of transect lines (i.e., 26, 13, 7, and 4) and the external validation subset.

| Statistic | 26-Lines | 13-Lines | 7-Lines | 4-Lines | Validation Subset |
|---|---|---|---|---|---|
| Observations | 3906 | 2119 | 1088 | 559 | 400 |
| Minimum | 2.62 | 3.28 | 3.28 | 3.28 | 3.63 |
| Maximum | 26.25 | 26.25 | 26.25 | 26.25 | 25.31 |
| Mean | 9.58 | 9.63 | 9.72 | 9.68 | 9.62 |
| Median | 9.30 | 9.41 | 9.96 | 9.65 | 9.53 |
| Standard Deviation | 3.39 | 3.44 | 3.59 | 3.56 | 11.20 |
| Skewness | 0.68 | 0.73 | 0.65 | 0.95 | 3.35 |
| Kurtosis | 0.22 | 0.58 | 0.66 | 2.10 | 0.78 |

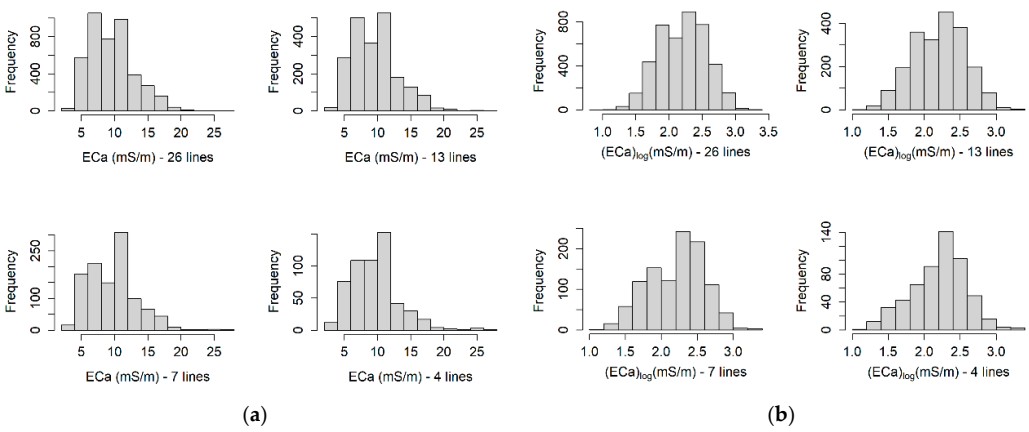

**Figure 4.** Histograms for soil ECa datasets for the different numbers of transect lines (i.e., 26, 13, 7, and 4), considering: (**a**) actual values; and (**b**) natural logarithm transformations.

### 3.1.2. Fitting Semivariogram Models

Spherical semivariogram model fittings were tested for anisotropy, with no significant RMSE differences for approach 1 cross-validations in the 30° and 120°; 40° and 130°; and 50° and 140° directions. Exponential and Gaussian model fittings were simulated, but did not improve RMSE in cross-validation. As a result, spherical models were used in all cases, as slightly larger ranges were suitable while searching for the maximum distance between transect lines.

Best fit variogram parameters for the natural logarithm of soil ECa datasets for the different numbers of transect lines are shown in Table 3. These parameters summarize best fit spherical models for all semivariograms for transect spacing simulations (Figure 5). Low nugget values were seen for ECa simulations with 26, 13, 7, and 4 transect lines, with respective line spacings of 40 m, 80 m, 150 m, and 300 m. That indicates a smaller variance in kriged maps and low standard deviation values. In spite of increasing distances between transect lines, nugget values were similar for seven or more transect lines, indicating that the distance between lines and, consequently, the reduction in the number of points contained in the ECa dataset affected the spatial dependence only for distances greater than 150 m between transects. As a metric of the impact of the nugget parameter on the final representation of the semivariance, the nugget parameter/sill ratio is shown in Table 3, column 5. Expressing this ratio value as a percentage allows a better comprehension and discussion about aleatory errors of semivariance. The ratio analysis suggests 150 m as the maximum distance between lines, since there is a clear influence of randomness in the semivariance in the four transect lines simulation. MCD values in this approach were between 184 and 192 m. Ranges varied from 498 to 530 m, representing the maximum distance beyond which there is significant loss of accuracy in the spatial pattern of ECa. All MCD are smaller than 300 m, and, according to [46], this measure can be considered a threshold distance that is proportional to the semi-variogram range and partial sill. Therefore, MCD may represent a maximum distance between transect lines. In this case, range parameter interpretations suggest that transect intervals around 150 to 190 m are within a safe threshold range. From the combination of nugget and range parameter indications, we would suggest no line intervals greater than 150 m.

**Table 3.** Best fit variogram parameters for the natural logarithm of soil ECa datasets for the different numbers of transect lines.

| Transect | Model | Nugget | Sill | Nugget/Sill (%) | Range (m) | MCD (m) |
|---|---|---|---|---|---|---|
| 26 | | $2.27 \times 10^{-3}$ | $1.48 \times 10^{-1}$ | 1.51 | 505 | 186.57 |
| 13 | Spherical | $2.76 \times 10^{-3}$ | $1.50 \times 10^{-1}$ | 1.81 | 521 | 191.72 |
| 7 | | $2.39 \times 10^{-3}$ | $1.62 \times 10^{-1}$ | 1.45 | 498 | 184.08 |
| 4 | | $7.00 \times 10^{-3}$ | $1.30 \times 10^{-1}$ | 5.11 | 530 | 188.59 |

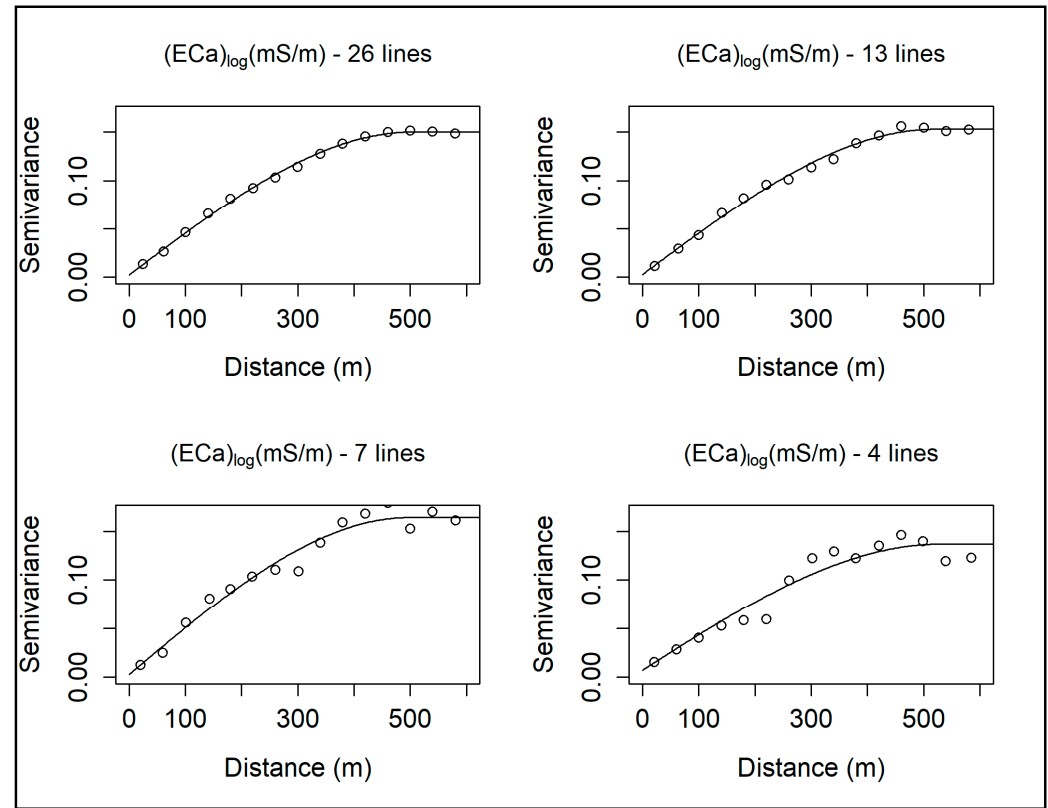

**Figure 5.** Empirical (circles) and adjusted (lines) variogram models for natural logarithm of soil ECa datasets from different numbers of transect lines (i.e., 26, 13, 7, and 4).

### 3.1.3. Mapping Soil ECa Spatial Variations

The soil ECa output map from 26 total survey lines is considered as a baseline for comparison against other transect spacing interpolations. High ECa values are located in the southwest portion of the map (Figure 6a), where a drainage channel can be observed. Although the EM38 survey was done in the dry season (July), on no till straw soil cover and with no irrigation regime, high ECa values may be associated with higher soil moisture and clay concentrations in this area, as reported by [3,10,34]. In contrast, lower ECa values in the northern part of the study area suggests a well-drained region with higher elevation and flatter sandy soils. These results match previous work from [35,47], where strong positive correlations were found between ECa values and clay and moisture concentrations.

For transect spacings equal to or greater than 80 m (13- and 7-line datasets), smoother ECa distribution patterns started to be observed, indicative of a threshold for sample reduction. Although this smoothing increases gradually from the 13- to 7-line dataset, it is still possible to observe the distribution of small-scale ECa features in the maps using the groups with 13 and 7 lines, when compared to the maps using the 26-line dataset. In particular for transect spacings larger than 150 m (4-line dataset), patterns of spatial structure have been clearly smoothed. The ECa maps interpolated from 13 and 7 lines (Figure 6b,c) display similar distribution patterns to the baseline map. The northern part of the 7-line output map shows a slight separation in a homogeneous area classified as low ECa values in the baseline map. Higher values are in the southeastern part, being smoothed at areas of low ECa values with more generalized patterns. However, the southwestern drainage can still be seen with a 150 m distance between lines.

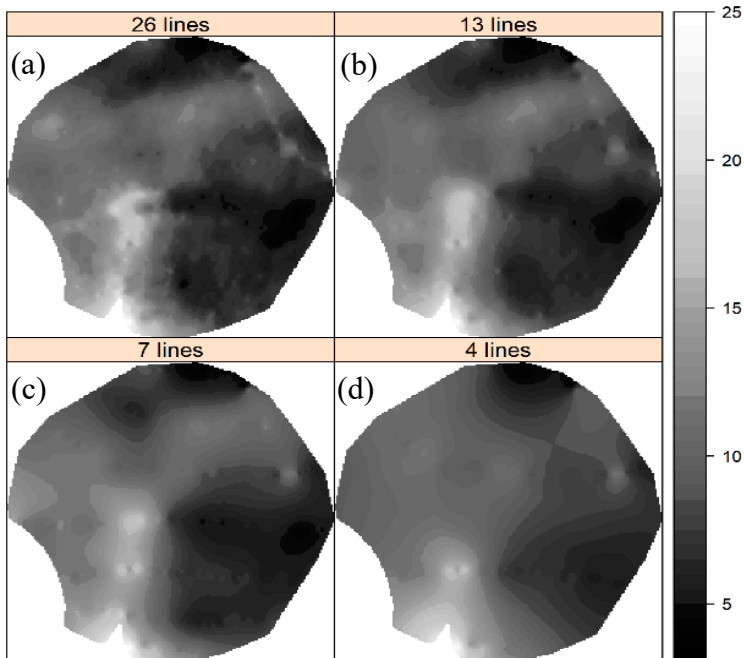

**Figure 6.** Soil ECa output maps from different number of lines, thus transect spacings of: 40 m (**a**—26 lines); 80 m (**b**—13 lines); 150 m (**c**—7 lines); and 300 m (**d**—4 lines).

Even though the 4-line output map still showed lower and higher ECa values corresponding with regions mapped with 26-, 13-, and 7-line interpolations, transect spacings of 300 m are higher than the MCD values observed in this approach (184 to 191 m). Even though there is pattern generalization and information loss in this interpolation, it is still possible to identify the drainage channel, although in shorter length if compared to the baseline map.

### 3.1.4. Map Uncertainty Assessment

Soil ECa map uncertainties based on external validation have RMSE and ME index values of 0.54 and 0.00, respectively, for the 26-line interpolation (Table 4). As expected, RMSE and ME values increase as the number of lines decreases, due to increasing uncertainty with smaller and scattered datasets. It can be observed that both ME and RMSE indexes are very responsive, showing differences in their values when comparing results from 26- and 13-line interpolations. These two datasets showed ME uncertainties values close to zero, where a zero ME value indicates unbiased simulations. ME from output maps using 13 and 7 lines have shown no significant difference in values in general, indexes show smaller values at finer resolutions, or more transect lines.

**Table 4.** Soil ECa map accuracy results, showing mean error (ME) and root mean square error (RMSE) from external validation.

| Transect Lines | ME | RMSE |
|:---:|:---:|:---:|
| 26 | 0.00 | 0.54 |
| 13 | −0.11 | 0.67 |
| 7 | −0.13 | 0.94 |
| 4 | −0.25 | 1.73 |

### *3.2. Approach 2—Different Sample Densities*

### 3.2.1. Exploratory Data Analysis

Unlike the descriptive statistics results in approach 1, minimum and maximum values were not alike for all datasets. Mean and median values from subset selection algorithms were stable across

sample densities (Table 5). This indicates that, even with a significant reduction in the number of points, datasets selected by the random and the Douglas-Peucker algorithms still preserved similarities regarding average values. Standard deviation values were similar in all datasets. As the standard validation dataset is the same, those results presented in Table 2 are not repeated here.

**Table 5.** Descriptive statistics results for soil ECa datasets from the different sample density reductions (i.e., 25%, 50%, 75%, and 95%).

| Statistics | 25% | | 50% | | 75% | | 95% | |
|---|---|---|---|---|---|---|---|---|
| | Random | DP | Random | DP | Random | DP | Random | DP |
| Observations | 2930 | 2933 | 1953 | 1960 | 977 | 982 | 196 | 196 |
| Minimum | 2.62 | 3.44 | 3.48 | 3.48 | 3.48 | 3.48 | 3.48 | 3.48 |
| Maximum | 25.59 | 26.25 | 24.02 | 26.25 | 24.02 | 26.25 | 20.00 | 24.02 |
| Mean | 9.59 | 9.62 | 9.63 | 9.63 | 9.70 | 9.56 | 9.80 | 9.74 |
| Median | 9.30 | 9.26 | 9.34 | 9.26 | 9.41 | 9.06 | 9.59 | 9.47 |
| Std. Dev. | 3.41 | 3.42 | 3.44 | 3.47 | 3.43 | 3.52 | 3.29 | 3.56 |
| Skewness | 0.66 | 0,73 | 0.66 | 0.83 | 0.67 | 0.94 | 0.52 | 0.83 |
| Kurtosis | 0.12 | 0.34 | 0.06 | 0.62 | 0.13 | 1.04 | −0.43 | 0.69 |

Values are moderately asymmetric positive for all densities. These values indicate that most of the ECa measurements have small values or are located to the left of the arithmetic mean, promoting a longer histogram "tail" on the right (Figure 7a,c). The kurtosis coefficients for all densification sets showed values higher than 0 mS/m, indicating that there is a flattening in the data distribution pattern, and the classification is leptokurtic. Thus, as the asymmetry and kurtosis values strongly suggest that data distribution for all sample densities may not be considered normal, a natural logarithm transformation was applied before variogram fitting and kriging (Figure 7b,d).

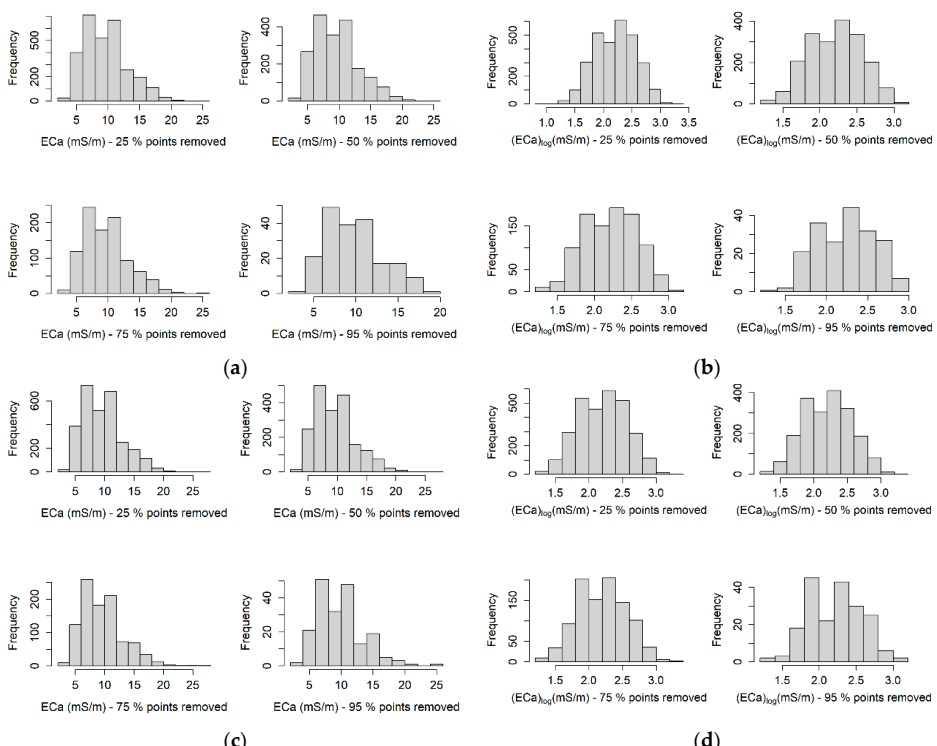

**Figure 7.** Histograms for soil ECa datasets from different sample density reductions (i.e., 25%, 50%, 75%, and 95%), considering: the random selection algorithm with actual values (**a**) and natural logarithm transformation (**b**); and the Douglas-Peucker algorithm with actual values (**c**) and natural logarithm transformation (**d**).

### 3.2.2. Fitting Semivariogram Models

Similar to approach 1, variogram simulations for anisotropy and exponential model fitting showed no significant differences of RMSE in the approach 2 cross-validations, suggesting that the larger range values from spherical isotropic fittings should be considered for maximum distance between observations. Variogram adjustment for all subsets from both selection algorithms had their best fit with spherical models (Figure 8; Figure 9). Nugget effects were small and of similar values, except for the 95% reduction subset from Douglas-Peucker (D-P) algorithm (Table 6). The D-P algorithm showed lower nugget values compared to random sampling, except for the 95% removal (Table 6). In the same way, range and sill values were similar for the two algorithms, except for 95%-point removal using D-P with slightly higher range value of 600 m (Table 6).

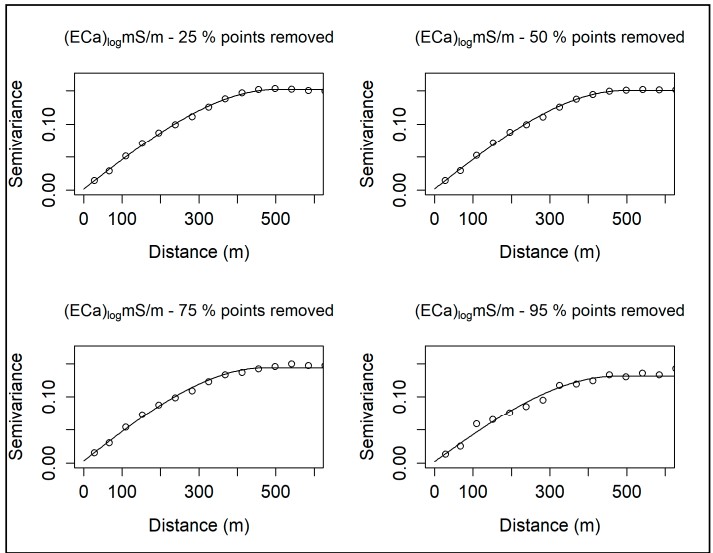

**Figure 8.** Empirical (circles) and adjusted (lines) semivariogram models for natural logarithm transformations of different soil ECa sample density reductions (i.e., 25%, 50%, 75%, and 95%), based on random point selection.

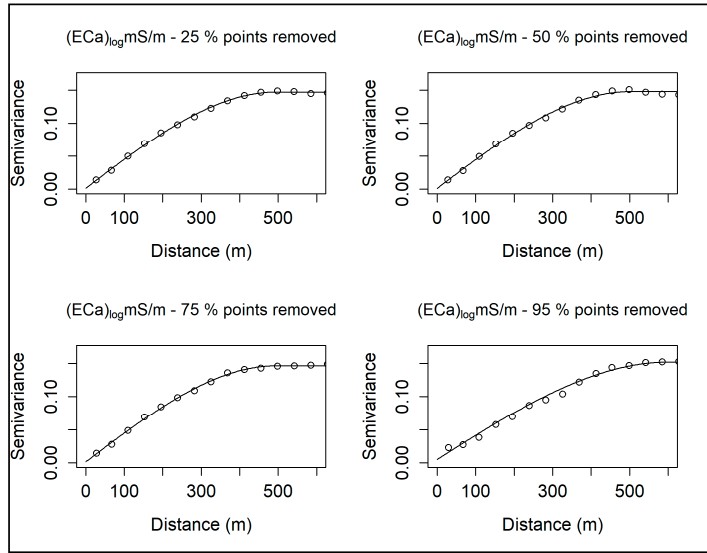

**Figure 9.** Empirical (circles) and adjusted (lines) semivariogram models for natural logarithm transformations of different soil ECa sample density reductions (i.e., 25%, 50%, 75%, and 95%), using the Douglas-Peucker algorithm.

**Table 6.** Best fit variogram parameters for the natural logarithm of soil ECa datasets, based on the different sample density reduction methods and levels.

| Density Reduction | Nugget | | Sill | | Nugget/Sill (%) | | Range (m) | | MCD | |
|---|---|---|---|---|---|---|---|---|---|---|
| | Random | D-P | Random | D-P | Random | D-P | Random | D-P | Random | D-P |
| 25% | $1.78 \times 10^{-3}$ | $1.44 \times 10^{-3}$ | $1.52 \times 10^{-1}$ | $1.47 \times 10^{-1}$ | 1.17 | 0.98 | 503 | 495 | 186.61 | 183.88 |
| 50% | $2.01 \times 10^{-3}$ | $1.35 \times 10^{-3}$ | $1.51 \times 10^{-1}$ | $1.48 \times 10^{-1}$ | 1.33 | 0.91 | 494 | 502 | 182.95 | 186.40 |
| 75% | $3.19 \times 10^{-3}$ | $1.26 \times 10^{-3}$ | $1.44 \times 10^{-1}$ | $1.46 \times 10^{-1}$ | 2.21 | 0.86 | 474 | 494 | 173.72 | 183.75 |
| 95% | $2.26 \times 10^{-3}$ | $5.00 \times 10^{-3}$ | $1.31 \times 10^{-1}$ | $1.52 \times 10^{-1}$ | 1.72 | 3.29 | 473 | 600 | 174.28 | 217.60 |

Similar behavior could be observed for the nugget/sill ratio, also indicating a stronger spatial structure of variation for D-P subsets as compared to random selections. Hence, even reducing the ECa samples from 2930 (25% removal) to 196 (95%), the nugget/sill ratio did not increase significantly (Table 6). When using the D-P algorithm, there was no increase in the uncertainty represented by the nugget/sill ratio up to the 95% removal level, and the ratio was smaller than for the random selection subsets, again except for the 95% removal level (Table 6).

MCD values were around 185 m, except for the 95% level of reduction by D-P, where there was a slight increase to 217 m (Table 6). These results are close to MCD values from the first approach, which ranged from around 184 to 191 m, indicating that a reasonable threshold for observation separation would be 150 m. The reduction in sample densities did not increase randomness in spatial dependencies, as no significant increase in nugget values was observed. This suggests that sample density reductions in this case could preserve the spatial structure of ECa variations.

### 3.2.3. Mapping Soil ECa Spatial Variations

Soil ECa output maps from the four dataset densities using the two algorithms captured the main spatial patterns of variation in the study area (Figure 10). High ECa values were in the center-southwest area, and low values located in the southeast and north parts of the area. These spatial representations are similar to the interpolated maps previously shown in approach 1. Even with a significant reduction in the number of sample points, the second approach illustrates high ECa levels in the central regions of the study area. The ECa maps from 95% point removal (196 points) were still able to illustrate similar visual results to ECa maps using 100% of points (3906 points; Figure 6a) using either random selection (Figure 10a) or D-P (Figure 10b).

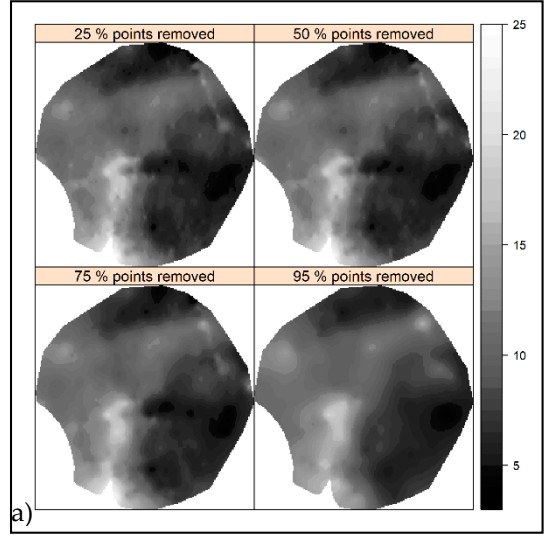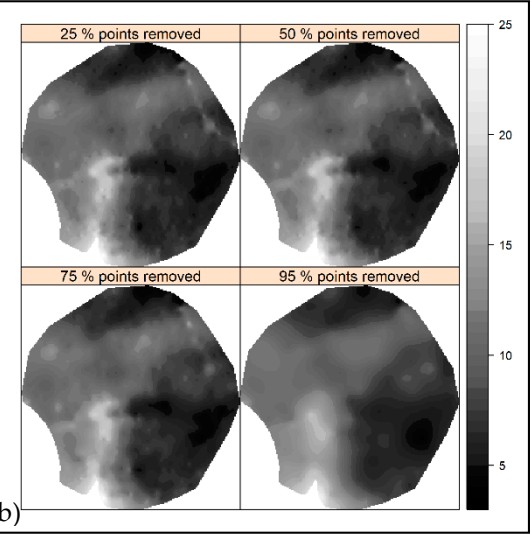

**Figure 10.** Soil ECa maps combining kriging interpolations of different sample density reductions (i.e., 25%, 50%, 75%, and 95%) selected by: (**a**) random algorithm; (**b**) Douglas-Peucker algorithm.

### 3.2.4. Map Uncertainty Assessment

Uncertainty index results in the line spacings approach are revisited here as a standard reference of 100% sample density (all points in all transects) against which to compare accuracies from the ECa maps of approach 2. Those results from the soil ECa map interpolated from 26 lines (Table 4, line 1, columns 2 and 3) were 0.54 for RMSE and 0.00 for ME. Overall results in the sample density approach include RMSE and ME values from map interpolations of all reduced datasets with an increasing trend as the total number of samples was reduced. Although these indexes are not exceptionally different between reduction algorithms, the D-P algorithm produced higher accuracy maps for all density subsets (Table 7), and the lower RMSE values from the D-P algorithm suggest a better spatial point distribution in ECa mapping.

**Table 7.** Soil ECa map accuracy results, showing ME and RMSE based on external validation for different density reductions with random selection and Douglas-Peucker algorithms.

| Density Reduction (%) | Random | | D-P | |
|:---:|:---:|:---:|:---:|:---:|
| | **ME** | **RMSE** | **ME** | **RMSE** |
| 25 | −0.01 | 0.6 | 0.01 | 0.56 |
| 50 | 0 | 0.64 | 0 | 0.57 |
| 75 | −0.01 | 0.72 | −0.01 | 0.66 |
| 95 | −0.05 | 1.11 | −0.08 | 1.04 |

Uncertainty indexes were low and similar for all subsets in the two selection algorithms, besides being very close to indices of the standard reference dataset up to 50% density reduction, in particular for the D-P algorithm. This suggested that if the number of samples was halved, accurate maps would still be produced. RMSE and ME values presented similar ranges, with small variations for sample densities reduced by 25%, 50%, and 75%, around 20% maximum uncertainty increase between all densities and resolutions, whereas the 95% sample density reduction simulations showed maximum uncertainty variation 35% above other results. All ECa sample density datasets (25%, 50%, 75%, and 95%), could represent major spatial patterns in soil ECa variations in the study area, when compared to the map produced from 100% sample density (Figure 6a).

Additionally, spatial differences in the standard deviation between ECa maps from the random selection and D-P algorithms are shown in Figure 11, as another way to visualize spatial uncertainty analysis. A map algebra procedure to subtract kriging results from both algorithms was used to map the standard deviation for their respective sample densities. Datasets with a 25% and 50% reduction, Figure 11a,b respectively, show minor differences in variances, while for 75% and 95%, Figure 11c,d respectively, differences in variances have increased slightly, with the highest variance difference in the maps with 95% of the points removed (Table 7). It can be observed that the Douglas-Peucker algorithm performs better at spatial point selection as compared to the random selection algorithm, as it better preserved map accuracy even with a 95% reduction in density.

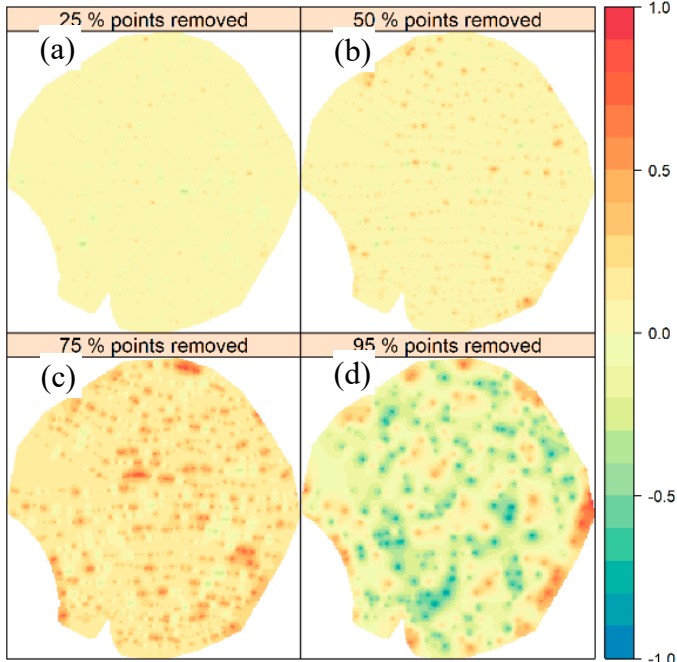

**Figure 11.** Differences between standard deviations of ECa from interpolations using random selection and Douglas-Peucker algorithms on different sample density reductions: 25% (**a**); 50% (**b**); 75% (**c**); and 95% (**d**).

*3.3. General Discussion*

Simulation datasets in the first approach, including 26, 13, and 7 transect lines, showed interpolation results using ordinary kriging that could represent main patterns of ECa variation in the study area. This supported the indication that future sampling designs should consider transect intervals from 100 to 150 m for ECa data collection on parallel back-and-forth paths. RMSE and ME values were higher when less than seven transects were included, suggesting that EM38 on-the-go survey paths should not exceed distances of 150 m between transect lines. Line intervals of 300 m (four lines) attenuated the representation of soil spatial variations in ECa maps, as RMSE and ME values from four lines interpolations were higher from external validation.

Evaluating the RMSE values between the two algorithms used to vary the density of points in the second approach, a reduction in the uncertainty values were found in all combinations between datasets of different sampling densities and spatial resolutions when the Douglas-Peucker algorithm was used in comparison to the random selection approach. The Douglas-Peucker algorithm provided values lower than those from the random selection algorithm for all combinations between sets of point density, except for the dataset with a 95%-point removal, corroborating the optimization by the D-P selection algorithm. Furthermore, in this algorithm RMSE values suggest a better spatial point distribution for ECa mapping procedures, in which the uncertainty range (from 0.56 to 1.04) suggests more accurate maps than for random reduction (from 0.60 to 1.11) and transect spacing (from 0.54 to 1.73) simulations.

Although soil ECa maps from EM38-MK2 devices have proven suitable supporting several correlated soil attributes, it is not possible to assure whether any soil process is actually occurring in the study area without pedotransfer functions been adjusted for ECa data as a function of other soil attributes data obtained via laboratory analysis.

**4. Conclusions**

The results of this study provide a basic step toward detailed investigations of variable ECa sampling densities. The use of the EMI sensors for precision agriculture applications in Brazil is

relatively recent. The dataset used in this work is among first data collection efforts throughout different regions and soil types. The following statements are addressed from this investigation:

- Sampling designs for continuous PSS surveys are still lacking optimal operational standards, potentially compromising map uncertainty evaluations;
- Datasets from different transect spacings and sampling densities could preserve similar ranges in the magnitude of soil ECa mapping uncertainty variations;
- Accurate soil ECa maps were obtained from increasing transect spacing simulations up to 150 m; or decreasing sample densities to a maximum 75% and limiting the distance between observations to 180 m.

**Author Contributions:** Conceptualization, G.M.V., H.M.R. and M.B.C.; Methodology, G.M.V., H.M.R. and M.B.C.; Formal analysis, H.M.R.; Investigation, G.M.V., H.M.R., R.P.O., S.R.L.T. and M.B.C.; Resources, H.M.R., G.M.V., R.P.O., S.R.L.T., M.B.C. and L.C.H.; Data curation, H.M.R. and G.M.V.; Writing—Original draft preparation, H.M.R.; Writing—Review & editing, H.M.R., G.M.V., R.P.O. and M.B.C.; Supervision, G.M.V. and M.B.C.; Project administration, L.C.H.; Funding acquisition, L.C.H., G.M.V. and R.P.O. All authors have read and agreed to the published version of the manuscript.

**Funding:** Empresa Brasileira de Pesquisa Agropecuária—Embrapa (Brazilian Agricultural Research Corporation) grant number 32.12.12.004.00.00, and Itaipu Binacional grant number 45.000.32.537.

**Acknowledgments:** The authors thank the Coordenação de Aperfeiçoamento de Pessoal de Nível Superior—CAPES (Brazilian Coordination for the Improvement of Higher Education Personnel) for the scholarship of Hugo M. Rodrigues, and the Associação do Sudoeste Paulista de Irrigantes e Plantio na Palha—ASPIPP (São Paulo Southwest Association of No-till Irrigation Farmers) and Agro Mario A.J. Van Den Broek Ltd. for their support.

**Conflicts of Interest:** The authors declare no conflict of interest.

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
