# Peer review of "Finding Suitable Transect Spacing and Sampling Designs for Accurate Soil ECa Mapping from EM38-MK2"

_soilsystems, doi:10.3390/soilsystems4030056_

Round 1
Reviewer 1 Report
This is an interesting work designed and developed with high rigor and with useful applications for improving the sampling campaigns.
One recommendation for the Approach 2 of the Sampling Design methodology. The proposed reduction of points (randomly) could be compared with a generalization method (i.e. Douglas-Peucker algorithm). In my opinion, your tests would be more complete and your conclusions more confident.
Some points that the authors should correct or better analyse:
You should try more attempts on the variogram fitting step. Some results could be very different when you use one model (spherical, exponential, etc...) or other, when you fit the chosen variogram to a local analysis at the first lags, or including mid or large distances. Additionally, different directional variogram analysis are need, absolutely for the present sampling based on transects where we can find some relevant directions (and poor sampling in the normal direction to the transect).
The variogram analysis is focused on the nugget and sill discussion, however I would expect some relevant contribution of range parameter to this discussion, for instance some discussion with the range obtained of the finest sampling (as indicator of spatial correlation threshold) and the spacing of the next samplings.
Some recommended references related: Oliver & Webster 1990, Kitanidis 1997, Pesquer et al. 2013.
Comments on specific lines of the manuscript:
Lines 125-128: Nowadays, time processing should not be a relevant issue for the dimensions of the study region and the range of explored spatial resolutions, excepting if you consider real time applications or very near-real time modelling.
Line 209: The use of global indicators (ME and RMSE) is interesting and useful. However, for a deeper spatial analysis, a regionalized indicator is needed, we need to add a spatial distribution of uncertainty. You can use local errors by the validation dataset and additionally the maps provided by the kriging interpolation methods (Lloyd & Atkinson 2001).
Line 240-241: The interpretation of nugget is approximately right but it could improve (referring to spatial uncorrelated error. Some recommended references (Oliver & Webster 1990, Kitanidis 1997, Pesquer 2013).
References:
Douglas D., Peucker T., 1973. Algorithms for the reduction of the number of points required to represent a digitized line or its caricature. The Canadian Cartographer 10(2), 112–122.
Kitanidis, P.K., 1997. Introduction to Geostatistics: Applications to Hydrogeology. Cambridge University Press.
Lloyd C.D, Atkinson P.M, 2001 Assessing uncertainty in estimates with ordinary and indicator kriging, Computers & Geosciences, Vol 27 (8) 929-937.
Oliver, M. A., Webster R., 1990. Kriging: a method of interpolation for geographical information systems. International Journal of Geographical Information Science 4(3), 313–332 pp.
Pesquer L., Pons X., Cortés A., Serral I., 2013. Spatial pattern alterations from JPEG2000 lossy compression of remote sensing images: Massive variogram analysis in High Performance Computing. Journal of Applied Remote Sensing 7 (1): 073595.
Author Response
Dear Reviewer.
We ask that you please consult the attached file that we have prepared very carefully to help you identify our responses and leading changes in the manuscript.

Reviewer 2 Report
The paper entitled “Fast and accurate soil ECa maps from EM38-MK2 2 survey: Finding suitable transect spacings and output 3 spatial resolutions” is an interesting paper presenting operational standards necessary for map uncertainty evaluations. Soil mapping is a valuable research tool in soil science, and I agree with this study’s objective that it is important to define mapping uncertainties related to sampling design. The findings of this study are useful for determination of ECa sampling designs.
L69: “laboratory analysis What may restrain” - A comma (,) is missing, small “w”, maybe “which” instead of “what”
L230: “pattern, which.” - something is missing here
Author Response

(The authors gave the same response as above.)

Reviewer 3 Report
This manuscript evaluated sampling design and spatial resolution influencing the quality of soil apparent electrical conductivity (ECa) maps. After reviewing the whole manuscript, I found there is a fundamental weakness that should be addressed before considering for publication:
The ECa based sampling design is normally based on the soil heterogeneity. The sampling points should be denser at high heterogeneous soil location and vice versa. Both methods proposed by authors that reduced the number of observed points from the original population in each transects or by enlarging distance line were not recommended. I speculate the readers may not benefit from the conclusions of this manuscript. Nevertheless, if the number of observed points was reduced according to the ECa pattern (soil heterogeneity), the analysis on the sampling design and spatial resolution may be interested to the readers. Moreover, the methods were tested only in one farm, where the data are insufficient to verify feasibility and applicability of the proposed methods.
Author Response

(The authors gave the same response as above.)

Reviewer 4 Report
While the reviewer sees the merit of an operational sampling procedure for Brasilian soil conditions the paper falls short on a number of issues.
Sampling design and EC38 operational issues have been discussed at length in the precision agriculture literature (e.g. see conference on precision agriculture). if this would have been reviewed before drafting the paper the focus would have been differently and worthwhile publishing. Reducing sampling density and it's influence on kriging has been studied in depth already.
Specific comments:
1)why ordinary kriging - what are the assumptions -
2) the authors comment several times on the time to run geospatial procedures - it's a matter of choosing the right software with good implementations. This is not a valid comment.
3)Discussion on pro/cons of EM38 missing - e.g. on some soils is just not providing enough evidence for spatial management decisions.
4) you have a pivot irrigation system in place with an elevation difference of 40m - quite impressive - how do you ensure water pressure difference due to that ?
5) where was the GPS and what type has been used ? RTK-GPS etc ?
6) while the paper used the usual space reduction, the time reduction has not been addressed properly (approach 2). a subsampling in R is one approach, but what about setting the time in the device or record every 5 record
7) what about replicability of the the data ?
8) why only omnidirectional SV - looking at the picture the reviewer would assume that the is a clear directional trend in it due to the a) landscape and b) due to the East -west track driven..
9) soil data in the field - what is the spatial variation in there ?
10) clear objectives before section 2 is missing
11) papers to look for EC38
Gebbers & Lueck e.g. quickly googled http://www.geo.uni-potsdam.de/mitarbeiterdetails/show/65/Erika_L%C3%BCck.html
https://www.researchgate.net/profile/Robin_Gebbers
on sampling designs and error: https://doi.org/10.1002/jpln.200521755
Author Response

(The authors gave the same response as above.)

Round 2
Reviewer 1 Report
Most of my previuos suggestions and concerns are rightly adressed in this new version. Specially, I'm happy with the inclusion of the Douglas-Peucker generalization method.
I don't agree with the authors' comment "Analyzing the experimental semivariogram, it is clear that the spherical model has the best fit". You should demostrate it. Most of the variograms (including exponential) can fit to the "usual" variogram form, we just select the right parameters and sometimes we can improve the fitting with the selection of the primitive function. Anyway, this is not a key point, this is a minor aspect. I agreee with the overall geostatistical approach applied in this work.
Author Response
Dear Reviewer
Please consult the attached document in word format with the answers to your considerations.
We thank you for your constructive and robust considerations, and we hope that we have reached the level of work required for the academic journal.

Reviewer 3 Report
I have no more comments.
Author Response

(The authors gave the same response as above.)
